# Tracking cryptic SARS-CoV-2 lineages detected in NYC wastewater

Davida S. Smyth [1,6], Monica Trujillo[2,6], Devon A. Gregory[3,6], Kristen Cheung[4], Anna Gao[4], Maddie Graham[3], Yue Guan[3], Caitlyn Guldenpfennig [3], Irene Hoxie[4], Sherin Kannoly[4], Nanami Kubota[4], Terri D. Lyddon[3], Michelle Markman[4], Clayton Rushford[3], Kaung Myat San[4], Geena Sompanya[1], Fabrizio Spagnolo[5], Reinier Suarez[3], Emma Teixeiro[3], Mark Daniels[3], Marc C. Johnson[3✉] & John J. Dennehy [4✉]

Tracking SARS-CoV-2 genetic diversity is strongly indicated because diversifying selection may lead to the emergence of novel variants resistant to naturally acquired or vaccine-induced immunity. To monitor New York City (NYC) for the presence of novel variants, we deep sequence most of the receptor binding domain coding sequence of the S protein of SARS-CoV-2 isolated from the New York City wastewater. Here we report detecting increasing frequencies of novel cryptic SARS-CoV-2 lineages not recognized in GISAID's EpiCoV database. These lineages contain mutations that had been rarely observed in clinical samples, including Q493K, Q498Y, E484A, and T572N and share many mutations with the Omicron variant of concern. Some of these mutations expand the tropism of SARS-CoV-2 pseudoviruses by allowing infection of cells expressing the human, mouse, or rat ACE2 receptor. Finally, pseudoviruses containing the spike amino acid sequence of these lineages were resistant to different classes of receptor binding domain neutralizing monoclonal antibodies. We offer several hypotheses for the anomalous presence of these lineages, including the possibility that these lineages are derived from unsampled human COVID-19 infections or that they indicate the presence of a non-human animal reservoir.

[1] Department of Life Sciences, Texas A&M University-San Antonio, San Antonio, TX 78224, USA. [2] Department of Biological Sciences and Geology, Queensborough Community College of The City University of New York, Queens, NY 11364, USA. [3] Department of Molecular Microbiology and Immunology, University of Missouri-School of Medicine, Columbia, MO 65212, USA. [4] Biology Department, Queens College and The Graduate Center of The City University of New York, Queens, NY 11367, USA. [5] Department of Biological & Environmental Sciences, Long Island University–Post, Greenvale, New York 11548, USA. [6] These authors contributed equally: Davida S. Smyth, Monica Trujillo, Devon A. Gregory. ✉email: marcjohnson@missouri.edu; john.dennehy@qc.cuny.edu

SARS-CoV-2 is shed in feces and can be detected by RT-qPCR in wastewater correlating to caseloads in sewersheds[1–3]. Consequently, municipalities and public health organizations have employed wastewater surveillance as a public health tool to make informed decisions about COVID-19 interventions[2,4]. However, the standard application of RT-qPCR does not provide genotype information and consequently cannot be used to monitor SARS-CoV-2 evolution and track variants of concern. Some researchers have applied, with mixed success, high-throughput sequencing strategies to total RNA extracted from wastewater. Often, coverage across the SARS-CoV-2 genome is uneven and epidemiologically informative regions can have low coverage[5,6]. Additionally, because wastewater samples contain an amalgamation of lineages circulating in the sewershed, it is not possible to reconstruct individual genomes using standard methods. Because of these difficulties, some researchers are using a strategy that employs the amplification and sequencing of small, specific regions of the SARS-CoV-2 genome, i.e., targeted sequencing[7,8]. Targeted sequencing can provide high coverage of epidemiologically informative regions of the genome and importantly, can reveal which polymorphisms are linked, thus allowing SARS-CoV-2 variants of concern (VOC) in communities to be tracked.

Since January of 2021, we sequenced SARS-CoV-2 RNA isolated from the raw influent from all 14 NYC WWTPs approximately twice per month[7]. Initially, we used an iSeq instrument to sequence a PCR-amplified region of the SARS-CoV-2 spike protein gene. This region spanned spike protein amino acid residues 434–505, which includes the receptor binding domain (RBD) (Fig. 1A). Beginning in April 2021, we switched to using a MiSeq instrument, which allowed us to sequence a larger amplicon that included amino acid residues 412–579. While no samples were analyzed with both the iSeq and MiSeq, the same constellations of mutations were consistently observed in the respective sewersheds regardless of the instrument used. These regions contain loci that are significant in SARS-CoV-2 receptor tropism and immune evasion, and contain multiple polymorphisms found in many VOCs[9,10].

## Results and discussion

**Identification of novel cryptic sewershed-specific lineages.** Our analysis pipeline, which uses the tool SAM Refiner to report polymorphisms and remove artificial chimeric sequences, allowed us to determine the frequency of each polymorphism and more importantly, elucidate which polymorphisms were derived from the same RNA sequence[8]. Freebayes and IGV were used to validate the reported polymorphisms (see "Methods" section). Using this approach, we were able to classify suites of mutations found in the RBD amplicons as consistent with Pango lineages B.1.1.7 (Alpha), B.1.351 (Beta), B.1.427/429 (Epsilon), B.1.526 (Iota), B.1.617 (Delta and Kappa), and P.1 (Gamma). Importantly, the distributions and trends in viral lineages from wastewater were consistent with patient derived sequences from NYC submitted to the GISAID EpiCoV database (hereafter, GISAID;

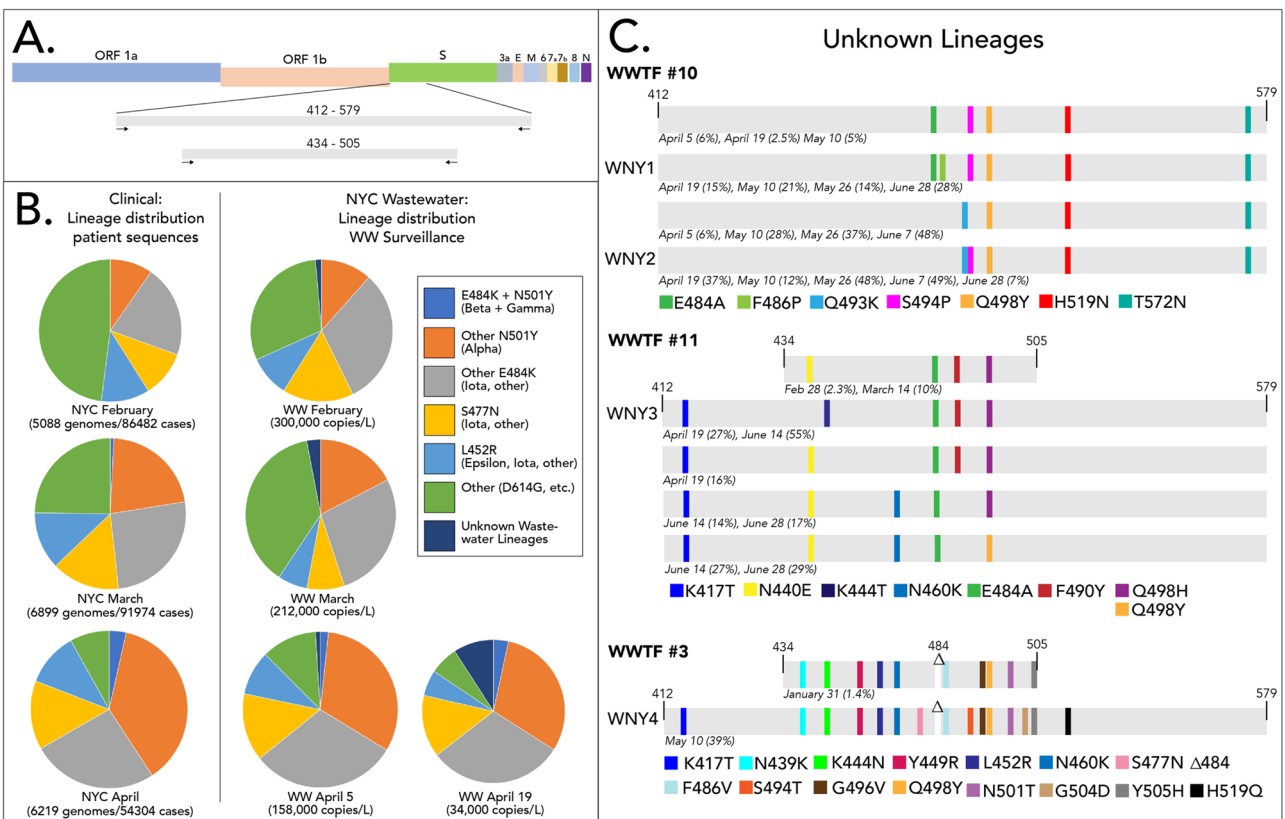

**Fig. 1 Novel SARS-CoV-2 lineages from wastewater. A** Schematic of SARS-CoV-2 and the amplification locations. **B** Distribution of SARS-COV-2 variants based on patient sequences (patient data obtained from GISAID) and wastewater surveillance. Polymorphisms detected from amplicon sequencing that were used to assign sequences to lineages are shown in the legend. The variants detected from the 14 NYC WWTPs were weighted by flowrate to generate a city-wide average distribution. **C** Novel lineages detected from WWTPs. Schematic highlights shared sequences identified from WWTP 10, 11, and 3 are shown. The percent of the sequences from each date that contained the indicated polymorphisms is shown below each lineage. The viral copies/ L corresponding to each date are shown in Supplementary Table 3. Some sequences have additional polymorphisms not listed. WNY lineage designations are shown for sequences used for tropism and antibody neutralization analysis. Source data are provided as a Source Data file.

https://www.gisaid.org/) (Fig. 1B and Supplementary Data 1 and Supplementary Fig. 1). For example, between February and April, wastewater and patient sequencing both revealed a notable increase in sequences assigned to the Alpha lineage and a corresponding decrease in sequences that did not belong to any of the VOC lineages.

In addition to well-recognized lineages, WWTPs 3, 10, and 11 contained RBD sequences with consistent constellations of polymorphisms detected over several months that did not match lineages reported in GISAID (Fig. 1C). Herein we refer to these constellations of linked mutations in the RBD sequences as lineages (meaning that they are of common descent), although without having the complete genome sequence we cannot say whether these were derived from a single lineage or multiple lineages with the same RBD sequence. These cryptic lineages were not static, as several of them appeared to acquire additional polymorphisms over the period of sampling. For example, one of the lineages from WWTF 10 added the polymorphism F486P at later sampling dates (Fig. 1C).

The cryptic lineages all remained relatively geographically constrained. The lineages from WWTP 3 and WWTP 10 were only observed from those locations during this sampling period. Sequences resembling the lineages from WWTP 11 were occasionally seen in neighboring sewersheds. Four of the anomalous lineages, designated WNY1, WNY2, WNY3, and WNY4, were selected for further study. Each of these lineages contained at least five polymorphisms; the most divergent was WNY4, which contained 16 amino acid changes in its RBD including a deletion at position 484. We note that WNY4 and the Omicron VOC possess mutations at the overlapping residues in the RBD, including K417, S477, T478, E484, G496, Q498, N501, and Y505. Polymorphisms at several of these positions have been reported to evade neutralization by particular antibodies[9,11–14].

Interestingly, all four WNY lineages contained a polymorphism at spike protein residue 498 (Q498H or Q498Y). As of November 30, 2021, there were only 35 SARS-CoV-2 sequences in GISAID that contained the polymorphism Q498H (eight in the USA), and none that contained Q498Y. However, both of these polymorphisms have been associated with host range expansion of SARS-CoV-2 into rodents[15–17], which are generally resistant to the parent SARS-CoV-2 lineage[18–20]. Notably, as the concentration of SARS-CoV-2 genetic material from NYC wastewater decreased along with the decrease in COVID patients, the fraction of the total sequences from these lineages has proportionally increased (Fig. 1C and Supplementary Fig. 1). By May and June, these lineages often represented the majority of sequences recovered from some sewersheds. For instance, on June 7 the sequences recovered from WWTF10 were predominantly composed of two variant lineages comprising 48 and 49% of the total sequences (Fig. 1C). By May, when cases were dramatically dropping, several of the NYC sewersheds did not contain high enough concentrations of SARS-CoV-2 RNA for analysis, which prevented further determination of city-wide variant distributions from wastewater.

As an external confirmation of our findings, we analyzed raw reads uploaded by September 15, 2021 to NCBI's Sequence Read Archive (SRA) from nearly 5000 other wastewater samples globally spanning 2020–2021, including 172 samples from New York state. Of all samples, only 7, all from NY state sewersheds, had sequences resembling the lineages we described (SRA Accessions: SRR15202279, SRR15384049, SRR15291304, SRR15128978, SRR15128983, SRR15202284, and SRR15202285).

**Are cryptic lineages derived from unsampled COVID-19 infections?** The existence of these cryptic lineages may point to COVID-19 infections of human patients that are not being sampled through standard clinical sequencing efforts. The frequency of weekly confirmed cases in NYC that were sequenced ranged from 2.6% on January 31, 2021 to 12.9% on June 12, 2021 (https://github.com/nychealth/coronavirus-data/blob/master/variants/cases-sequenced.csv). Nonetheless, not all cases were diagnosed and not all positive samples were sequenced. Therefore, it cannot be ruled out that the WNY lineages may be derived from patients, who are not being sampled in clinical settings.

Alternatively, these cryptic lineages may be derived from physically distinct locations in the body. That is, perhaps viruses of these lineages predominantly replicate in gut epithelial cells and are not present in the nasopharynx such that standard swabbing techniques can recover sufficient quantities for sequencing. Finally, we speculate that perhaps these mutations are found in minority variants that are unreported in consensus sequences uploaded to EpiCoV and other databases. Several groups have identified evidence of within host quasispecies in NGS datasets[21,22]. In one case, as many as 68% of the samples contained evidence of quasispecies in several loci, 76% of which contained nonsynonymous mutations concentrated in the S and orf1a genes[21]. To address whether our variants were associated with within-host diversity, we checked for minority variants in the raw reads of sequencing runs performed on samples collected between January 2020 to July 2021 obtained from NY state COVID-19 patients uploaded to the SRA. Of the 7309 samples publicly available as of July 21, 2021, none had sequences that matched the WNY lineages. Some sequences from these SRAs had subsets of mutations associated with the WNY lineages, but never a full suite or at a high frequency.

Arguing against the possibility of unsampled human strains is the geographical stratification of these cryptic lineages. Since January 2021, the lineages have remained geographically constrained over many months in the sewersheds we sampled, which is not consistent with a contagious human pathogen. While there were some COVID-19 related restrictions in NYC (e.g., restaurants operated at 50% capacity), movement was generally not restricted during the study period. Public transportation was operating in a normal capacity. Furthermore, our group regularly processes wastewater samples from over 100 locations and have never seen this kind of geographic constraint of a SARS-CoV-2 lineage that coincides with verified patient sequences. We suspect this lack of dispersal is consistent with infections of non-human animals with restricted movements or home ranges, but note that it could also be associated with patients confined to long-term healthcare facilities (e.g., nursing homes, hospices).

**Do cryptic lineages indicate presence of SARS-COV-2 animal reservoirs?** Another hypothesis is that these cryptic lineages are derived from SARS-CoV-2 animal reservoirs. To date, there have been a number of animals infected by SARS-CoV-2, including mink[23], lions and tigers[24], and cats and dogs[25,26]. To gain insight into the possible host range of these lineages, synthetic DNA coding for the amino acid sequences for these four lineages were generated and introduced into a SARS-CoV-2 spike expression construct for functional analysis (Fig. 2). All four of these lineages were found to be fully functional and produced transduction-competent lentiviral pseudoviruses with titers similar to the parent strain (D614G). To determine if these pseudoviruses displayed an expanded receptor tropism, stable cell lines expressing Human, Mouse, or Rat ACE2 were cultured with the pseudoviruses (Fig. 2). While the parent SARS-CoV-2 spike pseudoviruses could only transduce cells with human ACE2, all four of

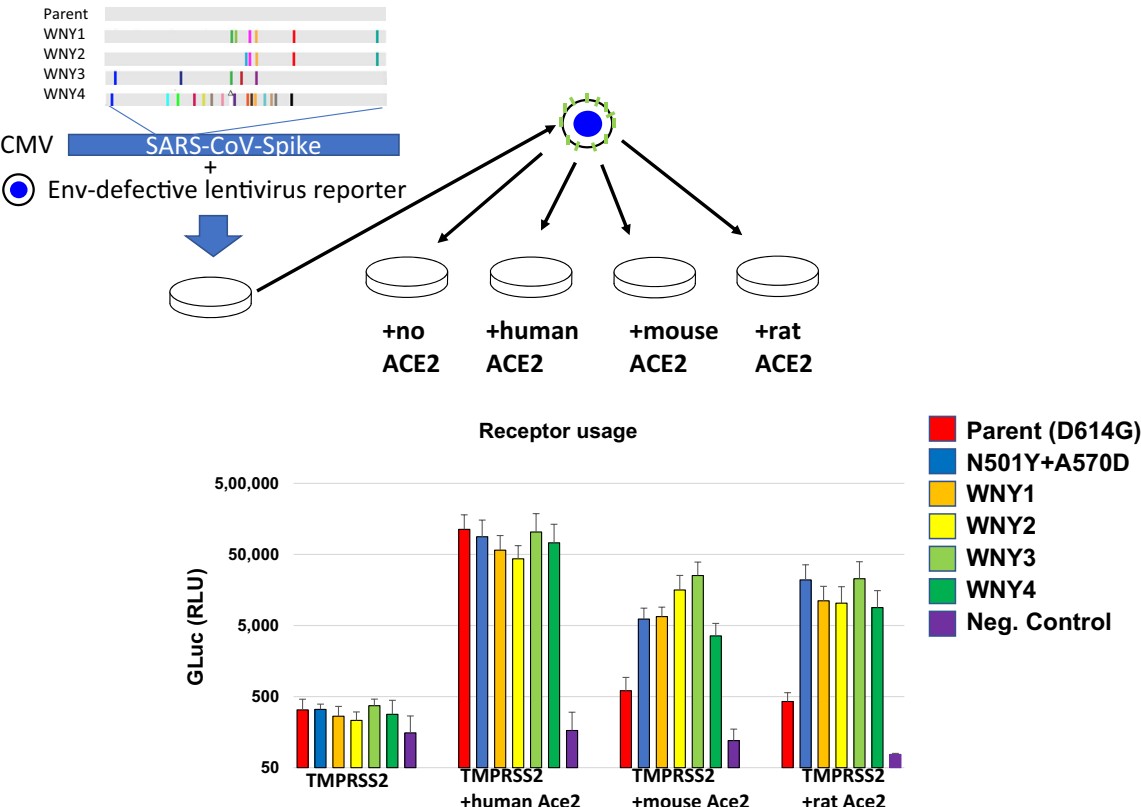

**Fig. 2 ACE2 usage by WNY lineages. A** Schematic of lineages and pseudovirion production. WNY1 = E484A/F486P/S494P/Q498Y/H519N/F572N, WNY2 = Q493K/S494P/Q498Y/H519N/T572N, WNY3 = K417T/K444T/E484A/F590Y/Q498H, WNY4 = K417T/N439K/K444N/Y449R/L452R/N460K/S477N/Δ484/F486V/S494T/G496V/Q498Y/N501T/G504D/505H/H519Q. The indicated mutations were introduced into a codon-optimized SARS-COV-2 expression construct. These constructs were used to produce lentiviral pseudovirions containing a *Gaussia* luciferase reporter. Pseudoviruses containing SARS-COV-2 Spike with N501Y/A570D were used as a control as this is known to be capable of infecting rodent cells. Pseudoviruses were used to transduce 293FT+TMRPSS2 stably transduced with human, mouse, or rat ACE2. The average and standard deviation from three independent experiments is shown. A two-way ANOVA revealed significant differences in receptor utilization ($F = 17.81$, DF = 3, 74; $P < 0.0001$). Source data are provided as a Source Data file.

the lineages could efficiently transduce cells with the human, mouse, and rat ACE2. Because some patient-derived SARS-CoV-2 lineages, such as Beta and Gamma, have also gained the ability to infect rodent cells (Fig. 2, N501Y+A570D), this observation cannot be taken as evidence that these lineages were derived from such a host[27]. Nonetheless, the observation is consistent with the possibility that these lineages are derived from an animal host such as a rodent.

If such reservoirs exist, the animal host would need to meet several criteria. First, the host species would need to be present in the sewershed. Second, the number of susceptible animals present must be high enough to sustain an epidemic for at least six months (i.e., the period for which we observe these sequences). Third, host animals must not disperse beyond the geographical locations where the sequences are found. Finally, there must be a route for shed viruses to enter the sewersheds where the lineages are seen.

We considered several mammalian species known to inhabit NYC that may meet these criteria, including bats (several species), cats (*Felis catus*), dogs (*Canis familiaris*), gray squirrels (*Sciurus carolinensis*), mice (*Mus musculus* or *Peromyscus leucopus*), opossums (*Didelphis virginiana*), rabbits (*Sylvilagus floridanus*), raccoons (*Procyon lotor*), rats (*Rattus norvegicus*), and skunks (*Mephitis mephitis*). To narrow our search, we reasoned that if viruses are being shed from one of these animals, then we should be able to detect rRNA from the animal in the sewershed as well.

**Mammalian species detected in wastewater**. We extracted total RNA from wastewater samples obtained on two different dates from sewersheds where the WNY lineages were observed. This RNA was PCR amplified with 12S rRNA primers (Supplementary Table 1) and deep sequenced. Sequences mapping to mammalian rRNA were observed in all samples (Table 1). In all cases, the majority of the rRNA sequences mapped to human rRNA. Several species, such as cow, pig, and sheep, were identified that are not indigenous to NYC. These detects are likely derived from food consumption so are ruled out as possible hosts. After non-indigenous mammals were removed, four remaining mammalian species were repeatedly detected: humans, cats, dogs, and rats (Table 1).

Cats and dogs are susceptible to SARS-CoV-2[28,29], and cats are able to transmit to other animals[26]. Many rodents are not permissive for infection by the canonical SARS-CoV-2 strain[20,30], but some variants have an expanded tropism that includes mice[27]. A 2013 census estimated that there are 576,000 pet cats in NYC households[31], but this estimate does not include stray cats. Extrapolating from a limited study conducted in 2017 implies a stray cat population of about 2500 animals[32], but this number does not accord with the approximately 18,000 animals received annually by NYC Animal Care Centers[31]. There are currently 345,727 active dog licenses in NYC[33], but this figure is likely a significant underestimate and the true number may be at least double this figure. Despite these uncertainties, both cat and dog

**Table 1 Predominant species detected in NYC wastewater via deep sequencing of 12S amplicons.**

| Genus | Common name | WWTP 3 6-7/6-28 | WWTP 10 6-7/6-28 | WWTP 11 6-7/6-28 |
|---|---|---|---|---|
| Homo | Human | M/M | M/M | M/M |
| Bos | Cow | ++/+++ | ++/++ | ++/++ |
| Sus | Pig | +++/+++ | +++/+++ | +++/+++ |
| Rattus | Rat | +++/+++ | +/− | ++/++ |
| Canis | Dog | ++/++ | +++/+++ | ++/+ |
| Felis | Cat | ++/++ | ++/+ | +/− |
| Ovis | Sheep | +/− | +/− | +/− |

Results from samples obtained on June 7 and June 28 are shown. The fraction of the total sequences detected are denoted as follows: Majority >50% (M), >1% (+++), >0.1% (++), <0.1% (+), not detected (−).

populations are dwarfed by the NYC rat population, which is estimated to number between 2–8 million animals[34].

WWTP 10 wastewater contained cat, rat, and dog rRNA, but rat rRNA reads were less than 0.1% of total reads and were only detected on one of the two dates tested (Table 1). This low detection was expected because the WWTP 10 sewershed is not a combined system (i.e., stormwater generally does not mix with wastewater). Moreover, the sewershed serves a suburban residential area and is believed to have one of the lowest rat densities in the city based on the volume of rat complaints received by city services (https://data.cityofnewyork.us/Social-Services/Rats-Heat-Map/g642-4e55). WWTP 3 and 11 wastewater also contained cat, rat, and dog rRNA, though the composition varied. In WWTP 3 wastewater, rat rRNA was the most prevalent after humans, representing over 1% of the total rRNA reads (Table 1). In WWTP 11, rat and dog rRNA were both above 0.1% of reads, but cat rRNA reads were less than 0.1% of total reads and were only detected on one of two dates tested (Table 1). All of these numbers are eclipsed by the overwhelming prevalence of human rRNA in the same samples. As no animal rRNAs are highly prevalent in all three sewersheds, it is difficult to reconcile a single animal being the reservoir for all cryptic lineages in NYC wastewater.

**Cryptic lineages detected from wastewater are resistant to some neutralizing antibodies.** In addition to polymorphisms from the cryptic lineages that are known to affect viral tropism, many of the polymorphisms are also known to affect antibody evasion. In particular, the WNY polymorphisms at positions K417, N439, N440, K444, L452, N460, E484, Q493, S494, and N501 have all been reported to evade neutralization by particular antibodies[9,11–14]. Most neutralizing antibodies against SARS-CoV-2 target the RBD of the spike protein, and most of these neutralizing antibodies are divided into three classes based on binding characteristics[35].

To test if the cryptic lineages have gained resistance to neutralizing antibodies, we obtained three clinically approved neutralizing monoclonal antibodies representing these 3 classes, LY-CoV016 (etesevimab, Class 1)[36], LY-CoV555 (bamlanivimab, Class 2)[37], and REGN10987 (imdevimab, Class 3)[38], and tested their ability to neutralize the cryptic lineages. All four of the lineages displayed complete resistance to LY-CoV555, despite the parent lineage remaining potently sensitive to this antibody (Fig. 3). WNY1 and WNY2 remained at least partially sensitive to LY-CoV016 and REGN10987, but WNY3 and WNY4 appeared to be completely resistant to all three neutralizing antibodies (Fig. 3).

Finally, we tested the ability of plasma from fully vaccinated individuals (Pfizer) or patients previously infected with SARS-CoV-2 to neutralize WNY3 and WNY4. All patients' plasma retained some capacity to neutralize these pseudoviruses (Fig. 3).

However, previously infected patients had an average 2-fold and 6.4-fold reduction in ID50 (WT vs. variant) with WNY3 and WNY4, respectively. Vaccinated patient plasma did not have a statically significant reduction with WNY3 but had an average 2.9-fold reduction with WNY4. It must be noted that neutralizing antibody activity from vaccinated individuals is not solely directed against the spike RBD. Therefore, if the full spike proteins from these cryptic lineages with the additional mutations they carry were tested, the neutralization capacity could be enhanced or further diminished. Thus, the characteristics of these lineages provide them the capacity to be a potential increased threat to human health.

**Challenges.** While we believe that our data, analysis, and interpretation of our findings warrant sharing with the scientific community, we recognize that our study has several limitations. The source of the novel lineages has not been identified. Investigations are ongoing to test possible animal reservoirs from these sewershed and to better pinpoint the geographical source of the cryptic variants by sequencing RNA from wastewater obtained upstream from our WWTPs of interest.

It is also recognized that the targeted sequencing approach does not identify mutations outside of the targeted region. In some cases, whole genome sequencing of wastewater has been employed, but the results have been ambiguous. Typical whole genome sequencing relies on amplification and subsequent computational assembly of genomes from overlapping 150–300 bp reads. When an infected individual's sample is sequenced, mutations appearing in different reads are assumed to be linked given that the reads likely come from a single virus genotype. By contrast, wastewater generally contains virions shed from numerous infected individuals, mutations identified cannot be reliably assigned to a specific genome[39]. To date it has not been possible to isolate viable virus from wastewater such that single virus genotypes can be sequenced[40]. Therefore, we cannot link mutations unless they are found on the same amplicon.

A further challenge is that the depth of coverage across the SARS-CoV-2 genomes sequenced from wastewater tends to be uneven. As such, phylogenetically and clinically important regions of the genome may fail to be adequately sequenced at appropriate levels of coverage. We chose to focus on a region of the spike RBD because of the prevalence of mutations that are phylogenetically and clinically important. We can reliably sequence this amplicon with high coverage.

To address the limitations presented by targeting just a small region of the SARS-CoV-2 spike, we are incorporating targeted sequencing of other variable regions of interest in the genome, particularly those regions that contain mutations unique to specific variants of concern. In addition, we are PCR amplifying, cloning, and sequencing a 1.5 kb region of the spike protein gene to confirm the linkage of mutations of interest.

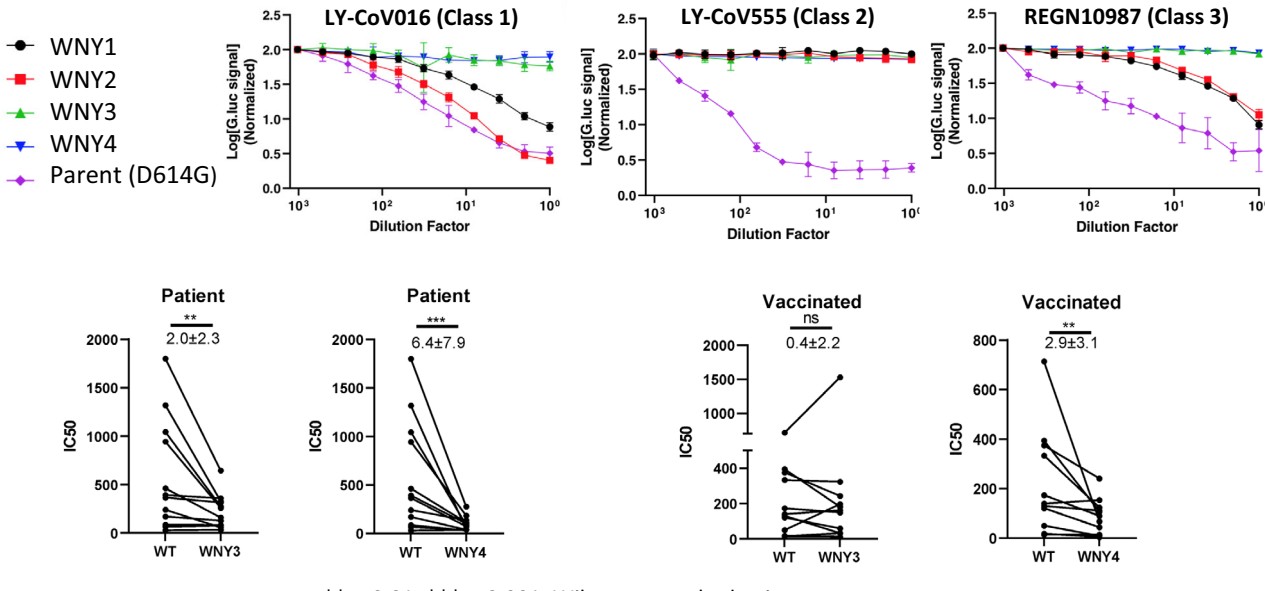

**p<0.01, ***p<0.001, Wilcoxon matched pairs test

**Fig. 3 Antibody resistance to monoclonal neutralizing antibodies and patient plasma.** Lentiviral reporter pseudoviruses containing *Gaussia* luciferase were generated with parent (D614G), WNY1, WNY2, WNY3, or WNY4 Spike proteins. These pseudoviruses were treated with 2-fold dilutions of indicated monoclonal neutralizing antibody or patient serum and used to infect 293FT+TMPRSS2+human ACE2. *Gaussia* luciferase levels were quantitated approximately 2–3 days of post-transduction. Representative examples of three experiments with monoclonal antibodies performed in triplicate are shown. Infection was normalized to the wells infected with pseudovirus alone. Patient plasma Neutralization IC50 titers were calculated using nonlinear regression (Inhibitor vs. normalized response—variable slope) in GraphPad Prism 9.0. The number indicates the mean fold of reduction in IC50 and SD. Wilcoxon matched-pairs signed rank tests, a two-tailed test, were performed for paired comparisons with significance levels as follows: WNY3 patient $p = 0.0049$, WNY4 patient $p = 0.001$, and WNY4 vaccinated $= 0.0068$. Source data are provided as a Source Data file.

| **Table 2 Demographic information for participants in antibody neutralization study.** | | | |
|---|---|---|---|
| | | **COVID-19 patients** | **Vaccinated** |
| Total | Number | 12 | 12 |
| Age | Average (Years) | 39 | 51 |
| | Range (Years) | 19–64 | 24–65 |
| Gender | Male | 1 | 3 |
| | Female | 11 | 9 |
| Vaccine type | | – | Pfizer |

**Summary.** To date, most data on SARS-CoV-2 genetic diversity has come from the sequencing of clinical samples, but such studies may suffer limitations due to biases, costs, and throughput. Here, we demonstrate the circulation of several cryptic lineages of SARS-CoV-2 in the NYC metropolitan area that have not been detected by standard clinical surveillance. While the origins of these cryptic lineages have not been determined, we have demonstrated that they have expanded receptor tropism which is consistent with expansion to an animal reservoir. Other SARS-CoV-2 animal reservoirs have been identified[23,41]. However, no single animal was strongly represented in all our rRNA sequencing analysis, which raises doubts that a single animal reservoir is the source of all the cryptic lineages.

Finally, we demonstrated that these cryptic lineages have gained significant resistance to some patient-derived neutralizing monoclonal antibodies. We note especially the high number of shared loci mutated in both our WNY lineages and the Omicron VOC. It's possible that these shared mutated loci are a product of convergent evolution to the shared selective pressure of antibody-mediated neutralization. Thus, these cryptic lineages could be relevant to public health and necessitate further study.

## Methods

**Ethics statement**. All procedures performed in studies involving human participants, including blood collection and processing, were approved by The Institutional Review Board of the University of Missouri (protocols #2043082 and 230262). Written consent was received from all human subjects prior to being enrolled in the study. The cohort of participants were selected based on equivalent levels of antibodies to SARS-CoV-2 RBD, age, or gender did not contribute to differences between samples (Table 2). COVID+ participants were collected prior to Dec 11, 2020 and did not receive a COVID vaccine. Vaccinated individuals were vaccinated with Pfizer and have not had a previous PCR+ COVID test. Patients were compensated $10/draw.

**Wastewater sample processing and RNA extraction**. Wastewater (24 h composite samples) was collected from the inflow at 14 NYC wastewater treatment plants and RNA isolated according to our previously published protocol[2]. While samples have been obtained and processed on a weekly basis since June 2020, we report herein the outcome of sequencing runs performed approximately 2 weeks between January and June 2021. The specific dates of sampling were January 31st, February 28th, March 14th, April 5th, April 19th, May 10th, May 26th, June 7th, June 14th, and June 28th.

Briefly, 250 mL from 24 h composite raw sewage samples obtained from NYC WWTPs were centrifuged at $5000 \times g$ for 10 min at 4 °C to pellet solids. Forty milliliter of supernatant was passed through a 0.22 µM filter (Millipore, SLGPR33R). Filtrate was stored at 4 °C for 24 h after adding 0.9 g sodium chloride (Fisher Scientific, BP358-10) and 4.0 g PEG 8000 (Fisher Scientific, BP233-1) then centrifuged at $12,000 \times g$ for 120 minutes at 4 °C to pellet the precipitate. The pellet was resuspended in 1.5 mL TRIzol (Fisher Scientific,15596026), and RNA was purified according to the manufacturer's instructions.

**Targeted PCR: iSeq sequencing**. RNA isolated from wastewater was used to generate cDNA using ProtoScript® II Reverse Transcriptase kit (New England Biolabs, M0368S). The RNA was incubated with an RBD specific primer (ccagatgattttacaggctgcg, Genewiz) and dNTPs (0.5 mM final concentration, included in the kit) at 65 °C for 5 min and placed on ice. The RT buffer, DTT (0.01 M final concentration, included in the kit), and the RT were added to the same tube and incubated at 42 °C for 2 h followed by 20 min at 65 °C to inactivate the enzyme. The RBD region was amplified using Q5® High-Fidelity DNA Polymerase (New England Biolabs, M0491S) using primers that incorporate Illumina adapters (see Supplementary Table 2). PCR performed as follows: 98 °C (0:30) + 40 cycles of [98 °C (0:05) + 53 °C (0:15) + 65 °C (1:00)] × 40 cycles + 65 °C (1:00).

The RBD amplicons were purified using AMPure XP beads (Beckman Coulter, A63881). Index PCR was performed using the Nextera DNA CD Indexes kit (Illumina, 20018707) with 2× KAPA HiFi HotStart ReadyMix (Roche, KK2601), and indexed PCR products purified using AMPure beads (Beckman Coulter, A63881). The indexed libraries were quantified using the Qubit 3.0 and Qubit dsDNA HS Assay Kit (Invitrogen, Q32854) and diluted in 10 mM Tris-HCl to a final concentration of approximately 0.3 ng/μL (1 nM). The libraries were pooled together and diluted to a final concentration of 50 pM. Before sequencing on an Illumina iSeq100, a 10% spike-in of 50 pM PhiX control v3 (Illumina, FC-110-3001) was added to the pooled library. The Illumina iSeq instrument was used to generate paired-end 150 base pair length reads.

**Targeted PCR: MiSeq sequencing.** The primary RBD RT-PCR was performed using the Superscript IV One-Step RT-PCR System (Thermo Fisher Scientific,12594100). Primary RT-PCR amplification was performed as follows: 25 °C (2:00) + 50 °C (20:00) + 95 °C (2:00) + [95 °C (0:15) + 55 °C (0:30) + 72 °C (1:00] × 25 cycles using the MiSeq primary PCR primers (Supplementary Table 1). Secondary PCR (25 μL) was performed on RBD amplifications using 5 μL of the primary PCR as template with MiSeq nested gene specific primers containing 5′ adapter sequences (Supplementary Table 1) (0.5 μM each), dNTPs (100 μM each) (New England Biolabs, N0447L) and Q5 DNA polymerase (New England Biolabs, M0541S). Secondary PCR amplification was performed as follows: 95 °C (2:00) + [95 °C (0:15) + 55 °C (0:30) + 72 °C (1:00)] × 20 cycles. A tertiary PCR (50 μL) was performed to add adapter sequences required for Illumina cluster generation with forward and reverse primers (0.2 μM each), dNTPs (200 μM each) (New England Biolabs, N0447L) and Phusion High-Fidelity DNA Polymerase (1U) (New England Biolabs, M0530L). PCR amplification was performed as follows: 98 °C (3:00) + [98 °C (0:15) + 50 °C (0:30) + 72 °C (0:30)] × 7 cycles +72 °C (7:00). Amplified product (10 μl) from each PCR reaction is combined and thoroughly mixed to make a single pool. Pooled amplicons were purified by addition of Axygen AxyPrep MagPCR Clean-up beads (Axygen, MAG-PCR-CL-50) in a 1.0 ratio to purify final amplicons. The final amplicon library pool was evaluated using the Agilent Fragment Analyzer automated electrophoresis system, quantified using the Qubit HS dsDNA assay (Invitrogen), and diluted according to Illumina's standard protocol. The Illumina MiSeq instrument was used to generate paired-end 300 base pair length reads. Adapter sequences were trimmed from output sequences using cutadapt.

**Wastewater rRNA sequencing.** cDNA from wastewater was also used to generate libraries using the primers indicated in Supplementary Table 1. rRNA Libraries were amplified using ProtoScript® II Reverse Transcriptase (New England Biolabs, M0368S) and pooled and sequenced on the iSeq100 as described above.

**Bioinformatics.** iSeq reads were uploaded to the BaseSpace Sequence Hub and demultiplexed using a FASTQ generation script. Reads were processed using the published Geneious workflows for preprocessing of NGS reads and assembly of SARS-CoV-2 amplicons[42]. Paired reads were trimmed, and the adapter sequences removed with the BBDuk plugin. Trimmed reads were aligned to the SARS-CoV-2 reference genome MN908947. Variants present at frequencies of 1% or above were called using the Annotate and Predict Find Variations/SNPs in Geneious and verified by using the V-PIPE SARS-CoV-2 application[43].

Reads from iSeq and MiSeq sequencing were processed as previously described[8]. Briefly, VSEARCH tools were used to merge paired reads and dereplicate sequences[44]. Dereplicated sequences from RBD amplicons were respectively mapped to the reference sequence of SARS-CoV-2 (NC_045512.2) spike ORF using Minimap2[45]. Mapped RBD amplicon sequences were then processed with SAM Refiner using the same spike sequence as a reference and the command line parameters "--alpha 1.8 --foldab 0.6". The output from SAM Refiner (available at https://github.com/degregory/Cryptic_WNY_sup/tree/main/SAM_Refiner_outputs) were reviewed to determine the known and novel lineage makeup of the sampled sewersheds. To verify and visualize the variant alleles, FreeBayes[46] was used to call variants on the mapped reads (https://github.com/degregory/Cryptic_WNY_sup/tree/main/VCFs) and Integrative Genomics Viewer[47] was used to generate genomic plots (https://github.com/degregory/Cryptic_WNY_sup/tree/main/IGV).

For sequencing from rRNA templates, dereplicated reads with a minimum unique count of 10 were mapped with Bowtie2[48] to a collected reference index of mitochondrial and rRNA related animal sequences from NCBI's nucleotide and refseq databases (https://www.ncbi.nlm.nih.gov/). Mapped rRNA sequences were reviewed for matching of specific organisms. Sequences with poor mapping to sequences in the index and a random selection of sequences with good mapping were checked by Blast (https://blast.ncbi.nlm.nih.gov/Blast.cgi) to verify the organism match. Matches were corrected based on the blast results as needed.

For both iSeq and MiSeq datasets, we examined Outbreak.info for the prevalence of each mutation and their associated lineages in New York, the United States and worldwide (Supplementary Data 1).

For sequences from GISAID, fasta formatted sequences from NYC patients were obtained from the GISAID database for submissions between January to April 2021. These sequences were processed similarly to the dereplicated sequences

above. Minimap2 was used to map the sequences to the spike ORF, then SAM Refiner was used to process the mapped sequences using "--min_count 1 --min_samp_abund 0" parameters to include all variations in the output.

Fastq formatted sequences were obtained for all sequenced SARS-CoV-2 clinical samples from New York state as of July 2, 2021, and all SARS-CoV-2 wastewater samples as of September 15, 2021. Metadata tables for all processed SRAs are available at https://github.com/degregory/Cryptic_WNY_sup/tree/main/SRAs. Fastq files were processed similarly to our iSeq and MiSeq sequencing runs with the merging step skipped for unpaired reads. Reads mapped to the spike Orf were processed with SAM Refiner with the parameters "--wgs 1 --min_count 1 --min_samp_abund 0".

**Plasmids.** Eukaryotic expression vectors for the heavy and light chains of antibodies LY-CoV016, LY-CoV555, and REGN10987 were obtained from Genscript. The lentiviral reporter constructed containing *Gaussia* luciferase (Gluc) with a reverse-intron (HIV-1-GLuc) was previously described[49]. The codon-optimized SARS-CoV-2 spike expression vector was obtained from Tom Gallagher[50]. This construct was modified to enhance transduction efficiency by truncating the last 19 amino acids and introducing the D614G amino acid change. DNA gBlocks containing the WNY RBD sequences were synthesized by IDT and introduced into the SARS-CoV-2 expression construct using In-Fusion cloning (Takara Bio, 638943). Lentiviral Mouse and Rat Ace2 vectors pscALPSpuro-MmACE2 (Mouse) and pscALPSpuro-RnACE2 (Rat) were obtained from Jeremy Luban[51].

**Cell culture.** The 293FT cell line was obtained from Invitrogen. The 293FT +TMPRSS2 and 293FT+TMPRSS2 + human Ace2 cells were previously described[52]. All cells were maintained in Dulbecco's modified Eagle's medium (DMEM, Cytiva, SH30022.01) supplemented with 10% fetal bovine serum, 2 mM L-glutamine (Sigma, G751), 1 mM sodium pyruvate (Sigma, S8636), 10 mM non-essential amino acids (Sigma, M7145), and 1% minimal essential medium (MEM) vitamins (Sigma, M6895). The ACE2 cell lines were generated by transfecting 293FT cells with 500 ng HIV GagPol expression vector, 400 ng of pscALPSpuro-MmACE2 (Mouse) or pscALPSpuro-RnACE2 (Rat), and 100 ng of VSV-G expression vector. Viral medium was used to transduce 293FT+TMPRSS2 cells[53], and cells were selected with puromycin (1 mg/mL) (Sigma, P8833) beginning 2 days postransduction and were maintained until control treated cells were all eliminated.

**Monoclonal antibody synthesis.** Transfections of 10 cm dishes of 293FT cells were performed with 5 μg each of heavy and light chain vectors and 40 μg polyethylenimine (PEI) (Polysciences, 23966-2)[53].

**Virus production and infectivity assays.** All transfections were performed in 10 cm dishes. 293FT cells were transfected with a total of 9 μg of HIV-1-GLuc, 1 mg of CMV spike vector, and 40 μg of PEI (Polysciences, 23966-2)[53]. Supernatants containing the virus were collected 2 days of post-transfection. Transduction of ACE2 expressing cells was performed by plating 30,000 cells in 96 well plates and co-culturing with 50 μL of HIV-1-GLuc/Spike particles. GLuc was measured 2 days post-transduction. All measurements were taken from distinct samples.

**Antibody neutralization assay.** Subjects were requested to provide a date of positive PCR test for SARS-CoV-2 and subsequently had laboratory-based serologic tests to confirm the presence of antibody against SARS-CoV-2 S1 RBD protein. A total of 10–20 mL of blood was collected from each participant. The plasma was then separated from the blood cells by centrifugation and stored at −80 °C.

**Pseudovirus neutralization assay.** All human plasma samples were heat inactivated for 30 min at 56 °C prior to the assay. Samples were diluted at 2-fold in ten serial dilutions in duplicates. Serially diluted samples were incubated with pre-titrated amounts of indicated pseudovirus at 37 °C for 1 h before addition of 293FT cells expressing human ACE2 and TMPRSS2 at 30,000 cells per well. Cells were incubated for 2 days and then the supernatant was used to measure *Gaussia* luciferase (RLU). All measurements were taken from distinct samples. Infection was normalized to the wells infected with pseudovirus alone.

**Statistical analysis.** Data and statistical analyses were performed in GraphPad Prism 9.0. A two-way ANOVA was performed to analyze the effect of receptor type and virus genotype on *Gaussia* luciferase intensity. Neutralization IC50 titers were calculated using nonlinear regression (inhibitor vs. normalized response—variable slope). Non-parametric pairwise analysis for neutralization titers were performed by Wilcoxon matched-pairs signed rank test.

**Reporting summary**. Further information on research design is available in the Nature Research Reporting Summary linked to this article.

## Data availability

Source data are provided with this paper. Raw sequencing reads are available in NCBI's Sequence Read Archive (SRA) under accession # PRJNA715712. Source data are provided with this paper.

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

## Acknowledgements

The research described herein would not be possible if not for the assistance and support of a wide-range of organizations and individuals that came together to address the shared calamity that is the COVID-19 pandemic. We thank Jasmijn Baaijens, Michael Baym, Gina Behnke, Esmeraldo Castro, Francoise Chauvin, Alexander Clare, Pilar Domingo-Calap, Robert Corrigan, Pam Elardo, Raul Gonzalez, Crystal Hepp, Catherine Hoar, Dimitrios Katehis, William Kelly, Samantha McBride, Hope McGibbon, Hilary Millar, Jason Munshi-South, Samantha Patinella, Krish Ramalingam, Andrea Silverman, Jasmin Torres, Arvind Varsani, Peter Williamsen, and members of the Dennehy Lab for support,

advice, discussions, and feedback. We also thank Molly Metz for assistance with graphics and figure design and Michael Loccisano with sample collection. This work was funded in part by the New York City Department of Environmental Protection, a donation from the Linda Markeloff Charitable Fund, and from the National Institutes of Health grant U01DA053893-01. The Water Research Foundation, the NSF Research Coordination Network for Wastewater Surveillance for SARS-CoV-2 and Qiagen Inc. provided resources, materials and supplies, technical support, and community support. Special thanks to Vincent Racaniello and the team at *This Week in Virology* podcast for connecting the New York and Missouri teams.

## Author contributions

M.T., D.S.S., M.C.J., M.D., and J.J.D. supervised the project. M.T., D.S.S., M.C.J., and J.J.D. conceptualized the project. M.T., S.K., D.S.S., M.C.J., M.D., and J.J.D. designed experiments. D.S.S., M.T., K.C., A.G., S.K., N.K., K.M.S., G.S., M.G., R.S., C.R., Y.G., and F.S. performed experiments. D.S.S., D.G., I.H., M.M., N.M., M.C.J., D.A.G., T.D.L., and J.J.D. performed data analysis and interpretation. M.T., D.S.S., D.A.G., M.C.J., and J.J.D. wrote the original and revised manuscript drafts. All authors contributed to reviewing and editing of the manuscript.

## Competing interests

The authors declare no competing interests.
