## [Peer Review File · Nature Communications]

Referee #1 (Remarks to the Author):

In this manuscript, "Tracking Cryptic SARS-CoV-2 Lineages Detected in NYC Wastewater," Smyth, Trujillo, Gregory and colleagues amplified regions of the SARS-CoV-2 Spike protein gene from RNA acquired from 14 NYC wastewater treatment plants (WWTPs) and examined the diversity of viral lineages with NGS. They report the potential detection and changing frequencies of novel SARS-CoV-2 lineages not recognized in GISAID's EpiCoV database, and they posit the possibility of a non-human animal reservoir. While the results are intriguing, some additional analyses and experiments would make the paper more solid and believable.

1) The authors used the tool SAM Refiner, but to really believe these variants, I'd like to see them appear in multiple variant calling tools and have variant allele frequency (VAF) profiles (they only say >1%), and IGV plots of each of them, especially since this is a rare tool for such an endeavor.

We have done this and can include the data in the manuscript.

2) They state that they used Bowtie2 or Minimap2 for alignment, but why did they use two different aligners? This is also unclear and might bias the results

This was done for technical reasons during development. The samples have been analyzed using both software packages and the results were the same. This can be clarified in the manuscript.

3) Were the alignments done requiring only unique matches?

No.

4) Did they see any differences in the samples on the iSeq vs. the MiSeq?

The sequences were equivalent with the same mutations being identified across both platforms as indicated in Figure 1. The technology is the same but differs in read length. The move from iSeq to MiSeq was done to increase sample throughput and coverage.

5) Of the 1,500 other global wastewater samples they examined, how many were from states close to NY? How many were in the US? It is possible that they just had a limited chance to see these Q498H or Q498Y variants due to lower sampling in these areas.

Since the paper submission, we have continued our analysis of available sequencing datasets uploaded to SRA. The only other wastewater samples with sequences even

vaguely related to our 'WNY' lineages were from NYC samples collected and sequenced by another group, Biobot.

6) They authors hypothesize that lack of dispersal is consistent with infections of non-human animals with restricted movements or home ranges, but humans also had restricted movements during this time, and this hypothesis should be considered as well.

We will consider it, but we regularly sample wastewater samples from over 100 locations and have never seen this kind of geographic constraint with a sequence that coincides with verified patient sequences. Also, NYC was not under lockdown and there were no restrictions on movement during the sampling period.

7) The authors only did 2 technical replicates for their 12S runs (Table 1), and rats were missing from WWTP10, which seems odd and unlikely.

That is inaccurate; we performed biological replicates in 2 different labs on 2 different dates. Furthermore, we are performing additional 12S analyses using a different protocol for isolating RNA from wastewater. As far as rat signal being missed from WWTP10, the sampling area is suburban and is known to have the lowest rat densities in the city.

8) In Figure 3, they show a lower IC50 in the antibody resistance to monoclonal neutralizing antibodies and patient plasma for the WNY3 and WNY4 variants, but there is one case where the IC50 went up, and a few times when it stayed flat. Given the low patient numbers, it is hard to confirm that these trends are solid and more replicates would help.

We tried not to overstate the result. We can repeat the experiment with serum from additional patients.

9) Could the authors validate some of these variants being carried in actual rats, cats, or dogs? That would be perhaps the best evidence of such a claim.

Yes, we are pursuing every avenue to do this, but these efforts are slowed by IACUC approval and city permit requirements. We are collaborating with the USDA APHIS to acquire wildlife samples including rats. However, these things take time, and we feel that it is reasonable and appropriate to share the data that we have now considering the potential implications.

Referee #2 (Remarks to the Author):

Smyth et al. present significant findings highlighting the strengths of SARS-CoV-2 S-gene sequence tracking through wastewater surveillance of samples from all NYC wastewater

treatment plants. The manuscript highlights the important role of SARS-CoV-2 sequencing from wastewater for pathogen surveillance. It also proposes that cryptic lineages or mutations are present in wastewater which have not been described from clinical samples. The studies further evaluated the potential contribution of some of these mutations to receptor tropism, suggesting binding to RBD, and possible resistance to monoclonal antibodies using a pseudovirus system. The authors offer multiple hypotheses for potential reservoirs of these cryptic lineages, which are not supported by experimental data and remain speculative.

Major comments:

1. The manuscript uses the term “lineages”, which is misleading as the work describes circulating S-gene RBD genotypes, not full-length genomes. While these regions cover many SNPs in VOC there is ongoing evidence for significant diversification of the SARS-CoV-2 genome, including other S-gene and non-S gene region of significance. There needs to be a clear distinction between the RBD data generated here and, whole-genome sequencing-based lineages. An alternative term should be considered, such as “S-gene RBD genotype”.

This can be amended.

2. Culture of the viruses representing these S-genotypes would significantly strengthen the study and provide more insights into the overall genome composition and phylogenetic relationship of these S-gene RBD genotypes with other known circulating lineages. While culture from wastewater has not been reliably established as a proof of concept see Bivins et al. (Environ. Sci. Technol. Lett. 2020, 7, 12, 937–942) provided evidence that infectious particles can be passaged in water and wastewater.

Virus cannot be passaged in wastewater; they are obligate intracellular parasites. Further, the Bivins et al. study did not show that infectious particles can reproduce in or be recovered from environmental samples. The study only showed that SARS-CoV-2 virions added to wastewater in the laboratory are rapidly degraded. We (and numerous other groups) have tried to culture virus from wastewater without success. Our results are available in a preprint accessible through the link below:

<https://www.medrxiv.org/content/10.1101/2021.07.19.21260777v1>

3. Additional sequencing information targeting the entire genome of samples positive for these S-gene alleles would also strengthen our understanding of the overall genetic background of these mutations.

Several groups have noted the challenges with obtaining sufficient coverage and depth of coverage of SARS-CoV-2 genomes from wastewater to assemble complete genomes. Illumina based technology results in short reads that must be assembled by comparison

with a reference genome. With a composite sample of wastewater, containing possibly thousands of viral genotypes combined with the issues of coverage and depth, assembling complete genomes is not likely. We have had varied success with WGS with our wastewater samples as well. We have chosen to use targeted sequencing as this allows us to get longer reads more reliably and with greater depth of coverage which each contain mutations along the same read associated with variants. Since we submitted our manuscript for review, we've been able to amplify longer regions (1.9kb) of the spike and confirmed colocalization of our novel mutations.

4. The Introductory paragraph should be expanded. Discussion of the difference between "standard" wastewater surveillance, aimed at quantifying SARS-CoV-2 titers using qRT-PCR, and the sequencing approach here should be included, including strengths and weaknesses of both.

This can be done.

5. Animal reservoirs are being proposed as a potential source of the described RBD genotypes. Additional data from trapped animals would strengthen this hypothesis, which if true would be an important finding.

We are trying. See above.

6. The sampling and sequencing occurred on the WWTP level. Additional data on sampling upstream collection sites (catchment area of these plants) would be highly informative to narrow down the potential sources of the genotypes.

Agreed, and we are actively trying this. However, these things take time, resources, and cooperation from other institutions, along with lengthy permission processes. We feel that it is reasonable and appropriate to share the data that we have now considering the potential implications.

7. Lines 98-99: "Therefore, the cryptic lineages may be derived from asymptomatic, vaccinated, immunosuppressed, pediatric, or chronically infected patients who are not being sampled in clinical settings." This statement appears entirely speculative, and it is unclear why these samples would be missed clinically. This could be addressed with targeted surveillance studies of individuals from areas sampled by the treatment plant.

While NYC is sequencing over 10% of its confirmed clinical cases, we wanted to acknowledge the possibility that these sequences are from patients that were overlooked for some unknown reason. However, 'targeted' surveillance of a sewershed with almost a

million people is not trivial and goes well beyond the scope of this study. Nevertheless, we are working to obtain fecal samples from patients dating from this period. In addition, we have confirmed that when generating a consensus sequence for submission to GISAID, our clinical colleagues are not including minority variants during this process. It is possible that our mutations are being discarded during the consensus generation process. However, as we state in the manuscript, we obtained raw sequence read files from 7,309 NYC clinical samples and did not see the anomalous lineages included as minority variants in any sample.

8. The analysis of 1500 publicly available WW sequence samples is included here. Was there any indication of the key mutations (not the 4 alleles focused on in here) in any of those samples?

Yes, we found an unusual variant in one other sewershed elsewhere in another part of the country after this manuscript was submitted. We are presently following up.

Additional comments:

1. Two different sequencing platforms generating different insert sizes were used to sequence the RBD regions; the rationale for using both approaches needs to be clarified.

We used an iSeq until we had access to a MiSeq. We can clarify this.

2. The data presented in the pie charts in Fig 1B are unclear; are the wastewater data based on averages across the 14 WWTPs? The provenance of the NYC clinical data should be specified as well (not only in the Methods section). As per the prior comment the use of "lineage" is misleading and should be further explained.

This can be clarified.

3. Line 66: How prevalent were these unknown wastewater lineages? Does "consistent but not static" mean that these were identified in many samples over time in these plants, but accumulated specific mutations? Please clarify further.

Yes, detected multiple times but not exactly the same each time. This can be clarified.

4. Further, while proportions of each RBD genotype are shown, the viral titer of wastewater samples is not described and would be useful in interpreting the relative prevalence of genotypes over time.

This can be added.

5. Line 78: “Notably, as the concentration of SARS-CoV-2 genetic material from NYC wastewater decreased along with the decrease in COVID patients...” The data supporting this observation appear to be missing; please show in either a primary or supplementary figure.

This can be done.

6. Line 81: “By May and June, these lineages often represented the majority of sequences recovered from some treatment facilities” The percentages displayed in Fig 1C do not seem to exceed 50% at any point.

Yes, they do. While no single lineage was above 50%, sometimes there were 2 or more lineages that added up to nearly 100% of the reads.

7. Lines 86-91: What types of facilities were these sequences from (if that info is available)?

The samples were obtained from the inflow of NYC wastewater treatment plants, but we are not permitted to say which ones.

8. Lines 114-115: Even though the within-host variation in clinical samples did not match the WNY lineages in their entirety, did these harbor any of the mutations found in the 4 WWTP lineages?

Some did, most did not. We can elaborate on this.

9. Lines 93-106: One alternative hypothesis is the continued mutation and replication of the virus during wastewater transport; is the reason for not suggesting this based on the idea that the virus is not likely to be actively replicating during transport?

Viruses are obligate intracellular parasites that can only replicate in a host cell. They absolutely cannot ‘replicate’ in wastewater.

10. Lines 116-121: Did any of the four lineages emerge in more than one WWTP?

Occasionally. The lineages from NY11 also appeared in another NYC sewershed a few times, but this was not mentioned in this manuscript. We tried to focus on the lineages which were the most consistent.

11. Figure 2 shows a “representative example of three experiments”. Could the authors present an average of all 3, including SD?

Yes.

12. Lines 173-177; Table 1: Table 1 needs more detail; what was the threshold for detection of mammalian rRNA? What were the relative levels of cat, dog, and rat rRNA in each of these sewersheds? Were these levels stable over time?

It was pretty cut and dry, but this can be clarified. We only did the experiment from two collection dates. We are now doing more analysis of mammalian rRNA. We have isolated total RNA/DNA from unfiltered and unpasteurized wastewater samples.

13. Lines 203-208: Too speculative that neutralization would be worse. Additional mutations in either S or elsewhere in the genome may impact the neutralization levels to go either way.

We meant mutations in the NTD which can block some neutralizing antibodies. We have most of the NTDs now and there are indeed many additional mutations that are known to block NTD neutralizing antibodies.

14. The discussion needs to be expanded and needs to have a paragraph on the limitations of the studies presented here.

This can be done.

15. Line 252: What read length was targeted (cycles used) for iSeq vs MiSeq sequencing?

MiSeq was 622, yielding 2x300 which was stated. iSeq is 2X150 bp.

16. In general, the workflow between sample processing and wastewater rRNA sequencing is not clear; perhaps a flow chart would be helpful in elucidating which samples were sequenced on the iSeq vs MiSeq and reasoning behind using these different approaches

Our workflows are now published. We can comment on the reasoning and clarify the different approaches used.

17. The justification and output from SAM Refiner are unclear; please describe how this approach allows for the assignment of sequence fragments with variant mutations to the same lineage, and how robust or accurate this approach is.

A manuscript describing SAM Refiner is now published (<https://www.mdpi.com/1999-4915/13/8/1647>). We can comment on the reasoning.

18. The figure legends need to be significantly updated to provide more detailed information; it is hard to follow many of the panels in relation to the text.

This can be done.

Referee #3 (Remarks to the Author):

This work describes wastewater-based epidemiology for analyzing markers of potential novel SARS-CoV-2 lineages in NYC treatment plants. A major limitation is the focus on only the Spike protein. Why obviously important for host tropism, it only partially explains variants/lineages of interest. For example, B.1.617.2 is explained by SNPs in ORF3a, M, ORF7a, and N in addition to SNPs on the S protein.

Much of the results and the discussion are speculative. The authors identified "novel lineages" (WNY1-4) that were not present in GISAID. The lineages did not disperse over time and the authors suggest it could be from a non-human animal reservoir. Mammalian rRNA was examined where the lineages was found with overlap from cats, dogs, and rats. Discussion of this analysis is purely speculative that includes references a "limited study conducted in 2017" of a stray cat population and the number of active dog licenses. The transfection/transduction work is noted but does not confirm much.

It is true that we do not yet know where these sequences are coming from, but we are actively working to identify the source.

The grammar in the manuscript could be improved, (pg. 6, line 116-117) as well as the figures (1c hard to read/visualize). There are also missing details such as read size, a map of WWTP locations, and clearer description of the sampling time frame.

We are not allowed to reveal the sewersheds, but we can provide the clarifications requested.

The authors need to add a limitations paragraph.

We can do this.

Peer review comments, further -

Reviewer #1 (Remarks to the Author):

In their manuscript Smyth et al summarize the main results of the application of an amplicon based method for the genomic surveillance of SARS-CoV-2 in NYC wastewaters. The authors report the identification in NYC wastewaters of several recurring mutations and haplotypes in the RDB domain of the spike glycoprotein, which are not commonly observed/ reported in currently available SARS-CoV-2 genomic sequences.

They propose some possible explanations for the origin/evolution of these novel variants, and suggest that sustained circulation in an alternative (from human) host in the sewer system might represent the most likely scenario. By performing affinity assays authors demonstrate that lentiviral pseudoviruses carrying a version of the SARS-CoV-2 spike glycoprotein with the NYC-sewers associated mutations have a higher affinity for rat and mouse ACE2, compared with the Wuhan strain of SARS-CoV-2.

Finally, they also show that the novel mutations they identify might also be associated with improved immune escape to some monoclonal antibodies.

In general the paper is well written, and the results are presented clearly. The topic is very interesting, however in my opinion the evidence presented by the authors is too circumstantial for a publication in a high impact factor journal in its current form.

Main points of concern criticism

It is not clear whether data obtained from the monitoring of waste-waters can recapitulate the prevalence/circulation of SARS-CoV-2 lineages in an accurate manner. For example while there is a good correlation, we can see from figure 1 that the estimates of the prevalence of lineages in NYC are markedly different when data from GISAID and data presented by the authors is compared

This is a relevant issue. I.e. authors need to prove that the method they apply are unbiased and reflect correctly the prevalence of different lineages

With their experimental design authors can only reconstruct a relatively small fragment of the Spike glycoprotein. It follows that all the experiments performed in the study are performed under the assumption that no additional mutations should be present in the rest of the protein/ genome. This assumption can not be verified and is unlikely.

Data provided by the authors is more qualitative than quantitative. Since the experimental design is based on PCR amplification of a target region, it is not clear if data at different time points can be compared

In Table 1 there is little evidence for the presence of rat rRNA in any of the locations sampled. Authors do not provide any likely/possible explanation. Human rRNA instead is highly abundant.

This would suggest a low concentration of rat fecal material in the sewers, and hence a low probability that viral RNAs identified therein identified should be of rat origin

No evidence of sustained circulation of SARS-CoV-2 in rats/mice has ever been reported, although the authors provide very interesting observations, these observations are based only on circumstantial data and speculations

COMMENTS ON POINTS RAISED BY ORIGINAL REFEREE 1

1) The authors used the tool SAM Refiner, but to really believe these variants, I'd like to see them appear in multiple variant calling tools and have variant allele frequency (VAF) profiles (they only say >1%), and IGV plots of each of them, especially since this is a rare tool for such an endeavor.

I agree only partially on this point: SAM Refiner was published only recently, and as such can not be considered the reference tool for its domain of application. However, I would not see why results provided by the tool should be questioned a priori. This said, independent validation of results by different/complementary approaches is a good practice that can actually improve the quality of the final results. So I would not discourage the application of other "more standard tools" to the same data, to provide a (sort of) independent validation

2) They state that they used Bowtie2 or Minimap2 for alignment, but why did they use two different aligners? This is also unclear and might bias the results

According to my understanding in the current version of the MS Bowtie2 was used for rRNA data, while Minimap2 for the analysis of the amplicon data. Since amplicon and rRNA sequencing data are not compared directly, this should not represent an issue in my opinion

3) Were the alignments done requiring only unique matches?

Does this question apply to rRNA data or to amplicon sequencing data? in the case of amplicon sequencing I suspect that this would not make much difference: the target region (RDB domain of spike) is known and does not include repetitive sequences

4) Did they see any differences in the samples on the iSeq vs. the MiSeq?

This is a good question. and should be explicitly addressed by the authors

5) Of the 1,500 other global wastewater samples they examined, how many were from states close to NY? How many were in the US? It is possible that they just had a limited chance to see these Q498H or Q498Y variants due to lower sampling in these areas.

This question in my opinion is now addressed properly. See lines 115-121

6) They authors hypothesize that lack of dispersal is consistent with infections of non-human animals with restricted movements or home ranges, but humans also had restricted movements during this time, and this hypothesis should be considered as well.

I partly disagree on this one. Even if the hypothesis raised by referee 1 is correct and should be addressed by the authors this in contrast with the observation that similar haplotypes/ sequences are not observed in the GISAID database in genomic sequences of viral isolates associated with the same area and interval of time

7) The authors only did 2 technical replicates for their 12S runs (Table 1), and rats were missing from WWTP10, which seems odd and unlikely.

I agree on this one. That was also one of my major points of concern. Authors should comment further and explain better

8) In Figure 3, they show a lower IC50 in the antibody resistance to monoclonal neutralizing antibodies and patient plasma for the WNY3 and WNY4 variants, but there is one case where the IC50 went up, and a few times when it stayed flat. Given the low patient numbers, it is hard to confirm that these trends are solid and more replicates would help.

I would prefer not to comment this one: it does not fall within my area of expertise

9) Could the authors validate some of these variants being carried in actual rats, cats, or dogs? That would be perhaps the best evidence of such a claim.

This is another very good point, and I raised a similar one myself. There is no evidence of sustained circulation of any SARS-CoV-2 variant in animal populations and/on in the possible hosts indicated in the study. Data provided by the authors on the enhanced affinity for mACE2 are purely "in vitro". Obviously if the hypothesis of sustained circulation of variants of SARS-CoV-2 in rats and/or other hosts could be proved, this would add a lot to the study

Reviewer #2 (Remarks to the Author):

I had previously reviewed this manuscript for Nature. A number of comments have been adequately addressed.

We are grateful for the reviewer's detailed and insightful comments on our manuscript titled, "Tracking Cryptic SARS-CoV-2 Lineages Detected in NYC Wastewater". We have also marked up an updated version of the manuscript with revisions aimed at addressing the issues highlighted by the reviewer. We also provide a point-by-point in response to the reviewer's comments in bold blue text below. We thank all reviewers and the editorial team for their time and consideration, which has significantly improved our manuscript.

REVIEWER COMMENTS

Reviewer #1 (Remarks to the Author):

In their manuscript Smyth et al summarize the main results of the application of an amplicon based method for the genomic surveillance of SARS-CoV-2 in NYC wastewaters. The authors report the identification in NYC wastewaters of several recurring mutations and haplotypes in the RDB domain of the spike glycoprotein, which are not commonly observed/reported in currently available SARS-CoV-2 genomic sequences.

They propose some possible explanations for the origin/evolution of these novel variants, and suggest that sustained circulation in an alternative (from human) host in the sewer system might represent the most likely scenario. By performing affinity assays authors demonstrate that lentiviral pseudoviruses carrying a version of the SARS-CoV-2 spike glycoprotein with the NYC-sewers associated mutations have a higher affinity for rat and mouse ACE2, compared with the Wuhan strain of SARS-CoV-2.

Finally, they also show that the novel mutations they identify might also be associated with improved immune escape to some monoclonal antibodies.

In general the paper is well written, and the results are presented clearly. The topic is very interesting, however in my opinion the evidence presented by the authors is too circumstantial for a publication in a high impact factor journal in its current form.

Main points of concern criticism

It is not clear whether data obtained from the monitoring of waste-waters can recapitulate the prevalence/circulation of SARS-CoV-2 lineages in an accurate manner. For example while there is a good correlation, we can see from figure 1 that the estimates of the prevalence of lineages in NYC are markedly different when data from GISAID and data presented by the authors is compared

This is a relevant issue. I.e. authors need to prove that the method they apply are unbiased and reflect correctly the prevalence of different lineages.

While we agree that the correlation between the virus prevalence in clinical samples and in the wastewater is important, we feel that it would be difficult to obtain a perfect correlation since the clinical sampling itself is biased. Generally, only 2-10% of positive clinical samples with a low Ct are sequenced in NYC. By contrast,

wastewater sequencing entails the random sequencing of virions secreted to the wastewater from the inhabitants of that area, thus can potentially more accurately identify the genotypes of infections producing asymptomatic/mild symptoms than can limited clinical sequencing. Indeed, our data show that clinical sequencing may be missing lineages observed in wastewater, but not in clinical sequencing.

With their experimental design authors can only reconstruct a relatively small fragment of the Spike glycoprotein. It follows that all the experiments performed in the study are performed under the assumption that no additional mutations should be present in the rest of the protein/genome. This assumption can not be verified and is unlikely.

We address this caveat in lines 257-261 and reiterate that the experiments only address binding affinity with the reconstructed fragment and does not consider the impact of additional mutations in the spike protein. We nevertheless feel this data is informative despite these limitations and we can cautiously extrapolate that the mutations do affect antibody neutralization.

Data provided by the authors is more qualitative than quantitative. Since the experimental design is based on PCR amplification of a target region, it is not clear if data at different time points can be compared.

Our main purpose is not to show changes in the prevalence over time, but rather to point out that these variants persist across long periods, pointing to the continued shedding of these variants into the wastewater of the respective areas.

In Table 1 there is little evidence for the presence of rat rRNA in any of the locations sampled. Authors do not provide any likely/possible explanation. Human rRNA instead is highly abundant.

We may be misinterpreting the reviewer's comment, but we feel that we've demonstrated the presence of rat rRNA in the wastewater of the sewersheds sampled. In some cases as high as 1-2% of the rRNA is from rats, which considering the disparity between rat and human body sizes, is a considerable amount. We elaborate on these findings in lines 230-233.

This would suggest a low concentration of rat fecal material in the sewers, and hence a low probability that viral RNAs identified therein identified should be of rat origin. No evidence of sustained circulation of SARS-CoV-2 in rats/mice has ever been reported, although the authors provide very interesting observations, these observations are based only on circumstantial data and speculations

We have toned down the language speculating that the unusual variants indicate a rat reservoir. We only wish to offer an alternative hypothesis for the presence of the unusual variants other than the possibility that they result from unsampled human

infections. SARS-CoV-2 animal reservoirs have been repeatedly demonstrated e.g. deer and mink.

COMMENTS ON POINTS RAISED BY ORIGINAL REFEREE 1

1) The authors used the tool SAM Refiner, but to really believe these variants, I'd like to see them appear in multiple variant calling tools and have variant allele frequency (VAF) profiles (they only say >1%), and IGV plots of each of them, especially since this is a rare tool for such an endeavor.

I agree only partially on this point: SAM Refiner was published only recently, and as such can not be considered the reference tool for its domain of application. However, I would not see why results provided by the tool should be questioned a priori. This said, independent validation of results by different/complementary approaches is a good practice that can actually improve the quality of the final results. So I would not discourage the application of other "more standard tools" to the same data, to provide a (sort of) independent validation

We have already described validation of the SAM refiner in the methods, where we used FreeBayes and IGV to validate and visualize the reported polymorphisms. We added additional text to the main text describing this validation using FreeBayes and IGV. Similar results were obtained from all analyses. See lines 75-76.

2) They state that they used Bowtie2 or Minimap2 for alignment, but why did they use two different aligners? This is also unclear and might bias the results

According to my understanding in the current version of the MS Bowtie2 was used for rRNA data, while Minimap2 for the analysis of the amplicon data. Since amplicon and rRNA sequencing data are not compared directly, this should not represent an issue in my opinion

We appreciate the reviewer's thoughts on this issue.

3) Were the alignments done requiring only unique matches?

Does this question apply to rRNA data or to amplicon sequencing data? in the case of amplicon sequencing I suspect that this would not make much difference: the target region (RDB domain of spike) is known and does not include repetitive sequences

We appreciate the reviewer's thoughts on this issue.

4) Did they see any differences in the samples on the iSeq vs. the MiSeq?

This is a good question. and should be explicitly addressed by the authors

Sequencing was performed on samples first on the iSeq and then on the MiSeq. Sequencing of the same samples was not done on both instruments. Despite this, the same suite/constellation of mutations was observed in the sewer sheds at different times. We specifically address this point on lines 65-67.

5) Of the 1,500 other global wastewater samples they examined, how many were from states close to NY? How many were in the US? It is possible that they just had a limited chance to see these Q498H or Q498Y variants due to lower sampling in these areas.

This question in my opinion is now addressed properly. See lines 115-121

We appreciate the reviewer's thoughts on this issue.

6) They authors hypothesize that lack of dispersal is consistent with infections of non-human animals with restricted movements or home ranges, but humans also had restricted movements during this time, and this hypothesis should be considered as well.

I partly disagree on this one. Even if the hypothesis raised by referee 1 is correct and should be addressed by the authors this in contrast with the observation that similar haplotypes/sequences are not observed in the GISAID database in genomic sequences of viral isolates associated with the same area and interval of time

We appreciate the reviewer's thoughts on this issue. We address the original reviewer's comments on lines 164-165 of the manuscript.

7) The authors only did 2 technical replicates for their 12S runs (Table 1), and rats were missing from WWTP10, which seems odd and unlikely.

I agree on this one. That was also one of my major points of concern. Authors should comment further and explain better

We address this non-detect on lines 219-233. Briefly, the samples were obtained from an area of the city that is known to have the lowest rat densities in the city. We feel the result we obtained is not surprising in this context.

8) In Figure 3, they show a lower IC50 in the antibody resistance to monoclonal neutralizing antibodies and patient plasma for the WNY3 and WNY4 variants, but there is one case where the IC50 went up, and a few times when it stayed flat. Given the low patient numbers, it is hard to confirm that these trends are solid and more replicates would help.

I would prefer not to comment this one: it does not fall within my area of expertise

9) Could the authors validate some of these variants being carried in actual rats, cats, or dogs? That would be perhaps the best evidence of such a claim.

This is another very good point, and I raised a similar one myself. There is no evidence of sustained circulation of any SARS-CoV-2 variant in animal populations and/on in the possible hosts indicated in the study. Data provided by the authors on the enhanced affinity for mACE2 are purely "in vitro". Obviously if the hypothesis of sustained circulation of variants of SARS-CoV-2 in rats and/or other hosts could be proved, this would add a lot to the study

We have been trying very hard to identify the source of these unusual variants. We have tested over 100 rat fecal samples from WWTP 10 and 11 and failed to detect the SARS-CoV-2 genetic signal. In addition, we are working with USDA/APHIS to examine rat derived samples from the same sewersheds. Despite not having a “smoking gun”, we strongly feel that this data should be reported, especially considering the emergence of the Omicron variant that shares many similar mutations with our cryptic lineages. We hope that our results will motivate continued surveillance for the emergence of novel variants of concern in wastewater.

Reviewer #2 (Remarks to the Author):

I had previously reviewed this manuscript for Nature. A number of comments have been adequately addressed.

We thank the reviewer for his detailed comments that significantly improved the manuscript.